# Recommendations for Measurement and Management of an Elite Athlete

**DOI:** 10.3390/sports7050105

**Published:** 2019-05-07

**Authors:** William Sands, Marco Cardinale, Jeni McNeal, Steven Murray, Christopher Sole, Jacob Reed, Nikos Apostolopoulos, Michael Stone

**Affiliations:** 1High Performance Department, USA Ski and Snowboard Association, Victory Ln, Park City, UT 84060, USA; 2Head Sports Physiology and Research, Aspire Academy, Doha POB 22287, Qatar; marco.cardinale@me.com; 3Department of Physical Education Health and Recreation, Eastern Washington University, Cheney, WA 99004, USA; jeni_mcneal@hotmail.com; 4Physical Education Program, University of California, Berkeley, CA 94720, USA; smurray@berkeley.edu; 5Department of Health and Human Performance, The Citadel–The Military College of South Carolina, 171 Moultrie Street Charleston, Charleston, SC 29409, USA; christopher.j.sole@gmail.com; 6Sport Science Department, WRC 121, University of Northern Iowa, Cedar Falls, IA 50613, USA; acob.reed@uni.edu; 7Kinesiology and Physical Education, University of Toronto, Toronto, ON M5S-2W6, Canada; nikos.apostolopoulos@utoronto.ca; 8Department of Sport, Exercise, and Recreation, East Tennessee State University, Johnson City, TN 37614, USA; stonem@etsu.edu

**Keywords:** single-subject research, trend analysis, statistical process control, elite athlete

## Abstract

Athletes who merit the title ‘elite’ are rare and differ both quantitatively and qualitatively from athletes of lower qualifications. Serving and studying elite athletes may demand non-traditional approaches. Research involving elite athletes suffers because of the typical nomothetic requirements for large sample sizes and other statistical assumptions that do not apply to this population. Ideographic research uses single-athlete study designs, trend analyses, and statistical process control. Single-athlete designs seek to measure differences in repeated measurements under prescribed conditions, and trend analyses may permit systematic monitoring and prediction of future outcomes. Statistical process control uses control charting and other methods from management systems to assess and modify training processes in near real-time. These methods bring assessment and process control into the real world of elite athletics.

## 1. Introduction

Studies of elite athletes are not rare, too often because the label “elite athlete” is applied too generously. Since 1974, the PUBMED.GOV online database holds records of a total of 6661 studies averaging 155 studies per year through 2018 discoverable with the search term “elite athlete.” The majority of studies of “elite athletes” have occurred since the year 2000 with a total of 5934, averaging approximately 312 studies per year (PUBMED.GOV accessed 6 April 2019). Measurement and management of athletes, elite or otherwise, is not a simple problem of categorization but one of acquiring reliable and valid data that can benefit all the stakeholders.

Traditional approaches to studies of sport and exercise science are built upon assumptions that rarely hold for the elite athlete. We believe that reframing elite athlete research and management will serve all of the stakeholders that pursue the highest levels of performance. The methods described below may also benefit any investigator, coach, or scientist who is faced with the problem of ensuring that training and performance are optimized for any single athlete.

### 1.1. What Is an Elite Athlete?

An elite athlete is someone who trains and competes at the highest levels of his or her chosen sport. Elite athletes are rare by definition. An elite athlete differs both quantitatively and qualitatively from athletes of middle- and lower-level qualifications. Although one can argue that elite athletes continue to learn throughout their careers, the elite athlete is not considered developmental nor competes in a developmental path or system. The elite athlete’s training and competition capacities are relatively stable, and his or her skills have reached an exceptionally high performance level [1]. Previously, the United States Olympic Committee defined an elite athlete as an athlete ranked among the top eight in the world [2]. Lorenz and colleagues [3] described an elite athlete as someone who is:“drafted or drafted in high rounds versus those undrafted or drafted in later rounds;perceived as having greater performance ability than that of their peers in the same sport;play at a higher level within a sport (division I vs II, professional vs amateur); andfor endurance, greater variables (e.g., running economy, AT, VO2max).” [3], p 542.

Elite athletes have backgrounds of great training volume, such as 7500 h in biathlon; 1.5 to 2 million ball strikes in golf; 150,000 to 600,000 arrows in archery; 2000 runs in bobsled, skeleton, and luge; and 250,000 to 500,000 elements per year in women’s gymnastics [4,5]. We will define an elite athlete as an athlete on a national team.

Too often, athletes described as elite are merely trained, highly trained, or included with a group of lower-level athletes. Studying elite athletes is difficult, and “estimates of enhancement of performance in laboratory or field tests in most previous studies may not apply to elite athletes in competitive events.” [6], p 472. In spite of recent controversies involving the role of practice in sports expertise [7,8,9,10,11,12]; the importance of many long hours of practice is still considered axiomatic to becoming an elite athlete. Lower-level and less-experienced athletes are comparatively deficient in training volume, experience, level of coaching, and talent [4,13]. 

### 1.2. The Covenant

Coaches and sport scientists working with elite athletes have a similar covenant to that of a physician and patient. While physicians are certainly directed by the results of experimental scientific research, documented trends, and other evidence-based-medicine; *they must cure or heal the patient in front of them* [14]. “Evidence based medicine (EBM) is the conscientious, explicit, judicious and reasonable use of modern, best-evidence in making decisions about the *care of individual patients*.” [15] p 219 (emphasis added). Physicians do not blindly apply a known method to a symptom profile and then consider their job complete. Physicians have to solve the individual patient’s problem, not some average patient. The sport scientist faces a similar covenant. The sport scientist must improve, stabilize, or repair the performance of the athlete he or she has, not the average athlete. The foregoing points lead to the use of individual athlete measurement and management (i.e., testing and analyzing a single athlete). 

### 1.3. Research on Elite Athletes’ Demands for Individualization

In order to determine if a meaningful change has occurred in some aspect of training and performance, the investigator needs to know the background variation of the variable(s) in question [16,17,18]. In other words, to know if something has moved to a new place one must know where it began, its direction of change, how much variation came from the athlete (biological variability), measurement methods, environment, and a variety of other factors [18,19,20]. 

Single-subject research offers a viable, if not preferable, paradigm for analyses of the elite athlete. If the reader has never used or heard of single-subject research, what we will refer to as single athlete assessment (SAA), and is only familiar with the methods of R.A. Fisher, Karl Pearson, and other statistical pioneers [21], then the idea of single-athlete research may cause cognitive dissonance [22,23]. However, one should appreciate that there are a number of noted scientific results that were based on an ‘N of 1.’ For example, Pavlov instrumented one dog [24], Broca dissected one brain [24], B.F. Skinner studied operant conditioning with only one athlete (i.e., animal) at a time [25], and Ebbinghaus studied memory using only himself as the subject [26]. Research methods involving a single athlete would seem to be tailored perfectly for the understanding and evidence-based support of elite athlete preparation and performance. 

An investigative question and a means of measurement in SAA approaches are usually determined before an experiment. SAAs are strong approaches to exploratory investigations often resulting in hypothesis generation rather than confirmation in the traditional research design sense [27,28]. One’s ability to be confident about measurements and what data might reveal are based on the use of multiple relevant measurements and similar results from different lines or types of inquiry. If multiple measurements of anything vary unreasonably, then one should be skeptical that whatever produced the measurements was inconsistent and may have arisen from multiple separate and independent or external causes rather than the treatment [29]. In group-based research, multiple measurements are conducted by virtue of doing the same measurements on many athletes. The variability of the measurements and their central tendency are characterized and then compared with the same measurements performed on a different group, the same group under different conditions, or both. Multiple measurements are performed with SAAs as well, but the measurements are performed on the same athlete. One could use SAA methods and simply accumulate a number of individual athletes and combine the data for typical nomothetic analyses, however such an approach may be more labor intensive and threaten internal validity. Concerns about the stability and repeatability of the measurements in both group and SAA investigations are shared [24,30].

Training monitoring approaches may share many common features with SAA investigations [31]. Monitoring training has become a field of sport science unto itself. Training diaries, whether by paper or computer, store long-term repeated measurements of an athlete. The dose–response information from training monitoring can be examined and can provide a historical context of the athlete’s development. Long-term collection of training data can serve as the raw material (i.e., archived data) for SAA methods. Both approaches intersect when measuring and managing the elite athlete. Monitoring and SAAs share repeated measurements and many aspects of analysis. However, monitoring may include group measurements and analyses, such as in team sports, and usually relies on collecting large datasets assuming that there is something worth measuring in the data. Data mining and “big data” methods are often employed in training and performance monitoring approaches [32,33]. SAAs share the same desires, but testable hypotheses are involved before data analyses. 

### 1.4. Case Study vs SAA

Common in the medical field, case studies and SAA are not the same but share some common features [24]. Case studies are descriptive in nature with no repeated measures of a baseline period (BP) (control or non-treatment condition) or a treatment phase or condition (TP) [34]. Clinical case studies make little attempt to reduce and/or eliminate rival hypotheses that may have influenced the outcome [34]. Case studies are a type of *one-and-done* approach to investigations, but can still be a useful research method for elite athletes, in particular when reporting training loads and/or specific training interventions. There is a paucity of real examples of the preparation of elite athletes in the literature, often for reasons of proprietary and competitive advantage. Documentation of what was used for training and how the elite athlete responded may still be helpful information for those who deal with elite athletes.

SAA methods are valuable in naturalistic investigations such as those associated with case studies and quasi-experiments involving observation and analyses of real-time and archived data (i.e., athlete monitoring). Quasi-experiments only approximate the demands of true experiments [35], and often have some aspect of the study that is out of the control of the investigator [36]. For example, during the baseline phase or the treatment phase of an athlete assessment, the athlete may be called away for personal, school, or family reasons. Either or both phases may be disrupted. This interruption may be of sufficient duration to compromise the integrity of either phase. One who is familiar with the practical circumstances of elite athlete training and performance will probably agree that nearly all of the potential investigations of elite athletes are or will be quasi-experiments. Kazdin describes the need for quasi-experiments as the middle ground lying between case studies and the clean, crisp elegance of controlled laboratory experiments [36]. Quasi-experiments, while not incorporating rigid laboratory controls, can provide reasonable evidence which may prove useful for good decisions in a messy world.

Given that the scope of SAA and training management is limited to one athlete, the investigator and training manager should be especially sensitive to the poor generalizability of SAA approaches to other athletes and settings [37]. Many of us trained by the methods of Fisher and Pearson have trouble embracing the idea that SAAs do not pretend to be generalizable to other circumstances. The results of studies with single athletes should not be amplified, upgraded, or over-interpreted for use with groups of athletes [30,38]. Those involved with the actual preparation and performance of elite athletes are rarely concerned with other athletes and situations as described below. 

### 1.5. Measurement for Management of the Elite Athlete–Why Nomothetic Approaches Can Fail

Nomothetic research designs (i.e., group-based analyses) are the most common method for assessing athletes of all levels by sports scientists and others. However, there is a problem in the application of typical scientific measurement and analysis practices with elite athletes. The rareness of elite athletes, unfortunately, makes finding groups of sufficient number that can be analyzed without violating numerous Fisherian statistical assumptions difficult or impossible [39,40]. A brief fictitious example may help emphasize the situation. 

Ten female national team athletes, involved in a strength and power sport, participated in a study investigating the effects of a new strength and power program for enhancing lower extremity explosiveness. With the chosen measure of change being a countermovement vertical jump (CMJ), these athletes, acting as their own control, performed a pretest CMJ for height. After completing the strength and power program for eight weeks, the athletes were retested, with the pre-test and post-test data shown in Table 1.

The results of a paired t-test were t_(9)_ = −2.29, *p* = 0.048. It appears that the strength program worked. Everyone should be happy. The average improvement was about 8%. However, note that the athletes with the highest pretest jumps (athletes 2, 5, and 8) actually got worse. Moreover, the coach knows that these three athletes are the only current hopes for podium finishes. Should the coach consider the intervention a success? Experience with dozens of presentations of this scenario to coaches has shown that not a single coach has ever considered this scenario a success. The three athletes who jumped poorly may have been the victims of other extenuating circumstances that accounted for their poor performance, but this further amplifies the point that without knowing the context of each athlete *individually*, using means and standard deviations may obscure information that is pertinent to the interpretation of this test and similar tests and treatments. 

### 1.6. Differences between the Baseline Phase and Treatment Phase

Repeated measurements are the foundation of SAAs. Use of a repeated series of BP measurements establishes an untreated control condition, followed by the same measurements imposed during and after the application of a TP condition. Thus, the athlete serves as his or her own control. SAA designs are often described as A-B designs, where A is the baseline phase, and B is the treatment phase. There are a variety of design approaches from A-B to A-B-A (baseline to treatment and then the treatment is removed for another baseline) to A-B-A-B (BP to TP to BP to TP), and so forth. If the treatment (B) measurements are sufficiently different from the baseline (A) measurements, then the treatment may have been responsible for the change. The protocols for SAAs may present data patterns such as those shown in Figure 1. 

### 1.7. Importance of the Baseline Phase

In order to detect a change, there must be an aspect of performance to which one can compare the new performance. A typical group study design assigns a treatment to one group of athletes but not the other group, and the comparison is performed between the two groups of athletes. In another group setting, a single group of athletes performs both the pre-test and the post-test using the pre-test scores as the baseline. Using the same athletes in each setting, pre-test, treatment, and post-test is known as a within-subjects (i.e., within-athletes) design. Within-athletes designs can be powerful because normal human variability is thought to be carried from the pre-test through the treatment and post-test, thus allowing the individual’s variability to be ‘subtracted out’ leaving only the influence of the treatment.

The importance of the BP in SAAs cannot be overstated. The BP serves as a descriptor of the current level of performance of the athlete. Baselines are the foundation from which predictions or comparisons are made, and therefore need to be stable [24,36]. The BP data should have relatively low variability, be without discernible trend, and be relatively free of error [36]. However, the requirements of baseline behavior are relative to the variable under assessment, the normal variability of the athlete, the timing of the baseline relative to treatment, and other factors [41,42]. 

Although having the BP prior to the TP is desirable, this is often not imperative. In some situations, the treatment may be the first phase needed immediately, leaving no time for observation to establish a baseline [36]. In other situations, the baseline behavior or performance may be obvious because the quality or magnitude of the treatment effects have never occurred before. In these circumstances, the design begins with the treatment or intervention in a B-A design. First, the treatment is applied (B), then the treatment is removed (i.e., shift to baseline, A). This approach may be most practical if the treatment is needed immediately to prevent injury or a serious learning or performance problem while still pursuing evidence-based assessment [36]. An example of this approach could be the first-time use of eyeshades or noise-abatement headphones on long plane flights with the outcomes being time and quality of sleep. 

## 2. Analysis of SAAs

Two primary approaches are used to analyze SAA data [36]. The first is visual inspection [24,36,41], with the second being statistical or quantitative procedures [30,41,43,44,45,46,47]. Alternatively, one can consider these two methods as the *order* in which SAA data are analyzed—first visually and then quantitatively. Visual inspection suffers from various aspects of bias [48] while offering rapid interpretation and classification of the obvious [49]. Pragmatically, coaches and athletes rarely take the necessary steps for statistical verification when making a choice about a training intervention or modality. However, everyone involved can benefit from evidence that rises above opinion, educated guess, rumor, and folklore. When an investigator pursues greater levels of confidence in a training decision, invoking appropriate statistical procedures can provide a higher degree of confidence [36]. Moreover, the methods of SAA analyses are simple and understandable to any coach.

Referring to Figure 1, the SAA data presents a decline in scaled peak force (SPF) of a men’s national gymnastics team athlete who showed a decline in SPF during weekly drop jump tests. The data recorded displays the means of two drop jump trials collected from 5 May to 18 September, with BP and TP collected from 5 May to 7 July, and from 17 July to 18 September, respectively. In early July (summer break), the coach abandoned specific conditioning for explosive jumping in favor of a more relaxed training approach that focused on skill and technique training. This change is believed to have been the catalyst for this particular athlete’s decrease in his fitness level for explosive jumping. The research question was: Should we regard the decline in explosive ability as large enough to have exceeded a chance occurrence and thereby lend some credence to the idea that the shift in conditioning could be responsible? We can reasonably assume that if the decline exceeds chance variation, then the coach has made a mistake that needs remediation. 

***Visual Inspection*.** Analysis of SAAs involves principles and methods similar to those found in nomothetic studies. Athlete training and management may benefit from simply graphing the data, with visual inspection being sufficient for determining whether a worthwhile effect or difference occurred [24]. The data show an obvious decline in SPF during the latter half of the period of interest. However, was the decline large enough to be unlikely to have occurred by chance? Sadly, what may seem obvious could fool the investigator [50].

***Quantitative Analysis.*** If the data are used for research purposes and/or there is a desire for increased confidence that the results are meaningful, then more rigorous analyses are merited. For example, a difference between BP and TP means of 1.5–2.0 times the BP standard deviation or the pooled standard deviation is considered indicative of a meaningful change based on typical error [51].

Following the approach described by Hopkins [52], the two periods shown in Figure 1 resulted in the following: The BP mean of SPF ± SD was 9.30 ± 0.62 N/N_BW_, CoV 6.7%, with the mean SPF value during the treatment phase being 8.20 ± 0.62 N/N_BW_, CoV = 7.6%. The difference between the BP and TP means was 1.14 N/N_BW_, exceeding −1.5 times the standard deviation of the BP and the pooled standard deviation 0.62 N/N_BW_. A value as large as or larger than −1.5 times the BP standard deviation or pooled standard deviation would only occur with a probability of approximately 7% based on normal probabilities. Since there are eight measurements in each phase, a paired t-test showed a statistically significant difference between the phases (t = 3.59_(7)_, *p* = 0.009, 95% CI = 0.39 to 1.89). In this case, the paired t-test was a procedure of convenience used to confirm further that there could be a worthwhile difference. The data appear to reveal an important change in the lower extremity fitness of this athlete and the coach should be informed of a potential need for a change in the conditioning program.

Before leaving the SAA methods, there are a litany of other approaches that can be used to assess phase differences in single-athlete data. Additional methods for the analysis of SAA phases include:percentage non-overlap of the data points in the BP compared to the TP (80% non-overlap is desired) [53],counting points above and below a celeration line can be used to detect phase differences [24,36], andconfidence intervals and effect sizes continue to be valuable and important [54,55,56], and others.

## 3. Analysis of Trends

The term ‘trend’ refers to a directional change. The previous section on differences in baseline and treatment phases may be described as a trend analysis, however these were relatively short-term and conclusions were binary, yes/no, different or not different. Trend analysis involves data that change systematically in relation to time. Athlete monitoring is probably the most common source of data in naturalistic or quasi-experimental trend analyses [31,57,58,59]. This section on trends refers to a longer duration of data collection as opposed to those commonly observed in phase difference assessments such as the A-B SAAs. Trends are distinguished by changes from which three patterns tend to emerge [47,60]: linear, a pattern of changes that can be best described by a straight line;curvilinear, a pattern of changes that is best described by a curved line, andcyclic, when data values repeat in a pattern.

Examples of these types of trends follow.

### 3.1. Linear Trend

Figure 2 shows examples of linear trends involving body mass and pre-practice resting heart rate (PPRH) from an elite female gymnast. Heart rate and body mass have been considered important monitoring tools to identify excessive fatigue and allostatic load [61,62,63,64]. Decreasing body mass could be indicative of protein catabolism and/or inadequate energy intake. A decreasing resting heart rate is indicative of enhanced fitness, while increasing resting heart rate may be indicative of increased metabolism and a stress response [65,66,67,68]. Moreover, decreasing mass in this elite female gymnast may also be indicative of enhanced fitness through pursuit of her competitive body mass [69]. The inflection-point or gap shown in Figure 2 from a decreasing PPRH trend to the increasing trend is coincident with the onset of illness symptoms (data not shown) in this particular athlete. 

***Visual Inspection.*** The linearity and slope of body mass as it changed over time are obvious. PPRH is more variable (i.e., noisier), but a downward followed by upward trend is also obvious, as shown by the plots of regression lines in Figure 2. 

***Quantitative Analysis.*** The line of best fit for body mass shows regression results as: (Y) = −0.05815(X) + 48.05(1)
where Y = predicted body mass, X = training day, −0.05815 = slope, 48.05 = Y intercept, R = −0.94, R^2^ = 0.89, and SEE = 0.24 kg.

One should be aware that each of the values in the regression equation has associated error terms. For example, the slope value standard error or deviation is 0.0142 kg; the constant value standard error or deviation is 0.3265 kg. These standard errors or deviations are rarely reported by the investigator, but one should appreciate that the values are not fixed but rather an estimate of a range of values [70]. 

The line of best fit for PPRH prior to day 20 showed the following: Y = −0.90827(X) + 74.73684(2)
where Y = predicted pre-practice resting heart rate (PPRH), X = training day, −0.90827 = slope, 74.73684 = Y intercept, R = −0.51, R^2^ = 0.26, and SEE = 9.23 bpm.

The linear equation shown indicates that the PPRH value was decreasing at a rate of about 1 bpm (−0.90827) on each successive training day. 

The line of best fit for the PPRH following day 20 resulted in the following: Y = 0.85614(X) + 40.0(3)
where Y = predicted pre-practice resting heart rate (PPRH), X = training day, 0.85614 = slope, 40.0 = Y intercept, R = 0.66, R^2^ = 0.43, and SEE = 5.70 bpm.

In contrast, during the period starting at day 20, the regression analysis showed an increase of approximately 1 bpm (i.e., 0.85614 bpm) per training day. The interpretation of the predicted PPRH values follows the same idea as those for mass described above. One should not overinterpret the predicted Y values of PPRH because the error is much larger than the change in beats per minute. The quantitative analysis supports the visual inspection by verifying the negative slope of body mass and a negative followed by a positive slope of the PPRH. Decreasing body mass coincident with increasing heart rate is a ‘classic’ symptom of excessive fatigue and overtraining [71,72]. In the case of the specific data presented here, the eventual outcome for this athlete was a career-ending injury on the last data point shown.

### 3.2. Curvilinear Trend

A curvilinear example is shown in Figure 3 involving the changes in a single male national team weightlifter’s hydration level over a period of approximately three and a half months. The hydration values were obtained weekly via urine specific gravity (USG) using a manual refractometer. 

***Visual Inspection.*** The USG data are plotted in Figure 3. If the measured USG value lies above 1.02, the athlete was considered dehydrated and counseled to increase fluid intake [73]. The long-term change in USG values depicts a partial U-shaped curve (Figure 3). The raw data alone are visually classified easily as curvilinear, and the USG values relative to the 1.02 boundary limit show when the athlete met a desired level of hydration. Potential explanations for the U-shaped curve are a change in motivation of the athlete or a shift from competition preparation and/or increased demands for body weight control shifting to less strict body weight demands during training.

***Quantitative Analysis***. The athlete’s USG values depict a curvilinear change over time. Although the curvilinear change in hydration is obvious, the primary practical judgment to be made from these values was whether the USG value was greater or less than 1.02 USG. The best curve fit of the USG data is shown by the red-dotted line which is calculated by a second order polynomial. Ninety-five percent prediction confidence limits are also shown. The regression information is shown below:Y = 0.0000250444(X^2^) – 0.00135974869(X) + 1.02518775(4)
where Y = predicted USG, X = training day, R = 0.65, R^2^ = 0.42, and SEE = 0.046.

Based on the results of the USG trends in Figure 3, the athlete may merit a warning that his USG is slipping toward the dehydration criteria of USG ≥ 1.02, if the trend continues. Using the regression equation, the athlete will likely reach the 1.20 threshold by the 51st day, or approximately 11 days from the last data point. The prediction is relatively weak (R^2^ = 0.42), but trends and regression analyses can help the sport scientist obtain more information about the seriousness of a trend and decide whether immediate action is necessary.

### 3.3. Cyclic Trend

Cyclic data refers to repeated wave-like behavior with rising and falling values. Cyclic trends usually require longer duration data sets, such as the three and a half months shown in Figure 4 [74,75,76]. The duration of data collection may have an influence on the ability to observe cyclic behavior. Rapidly undulating data such as neuromuscular data may show cyclic behavior in only a few seconds, while fitness data may require weeks. Stopping data collection prematurely may show only a portion of the cyclic behavior leading to an aliased or incomplete picture of the existing data pattern. 

***Visual Inspection*.** Training load is often cyclic based on a hard-day-easy-day approach and weekly accumulations of training load punctuated by rest days on weekends. The values in Figure 4 show the natural result of weekly training load variations with a periodicity of approximately five training days per week. Cyclic training loads such as the volume values in Figure 4 clearly show the up and down characteristics of a wave or an undulating pattern, as described in many training theory texts [77,78,79].

***Quantitative Analysis***. Figure 4 shows the raw volume data and two processed data sets using 2-point and 4-point running averages. A running average is often used to “smooth” noisy serial data in order to show overall change while retaining fidelity. Smoothing of serial data can be achieved with various calculations, but a running average is among the simplest. A running average is a simple average of two or more data values obtained via successive calculations or iterations through the entire data set (Figure 4). Note that the 2-point running average (red dashes) retains most of the features of the original dataset while the 4-point running average (blue dots) deviates considerably from the original data. A running average, along with other similar techniques, also tends to shift the dataset forward in time (to the right in Figure 4) [74,76,80]. If the data are “over-smoothed,” (i.e., 4-point running average in Figure 4), one risks aliasing the resulting dataset and losing the opportunity for a clear interpretation [31,43,74].

Note that the cyclic nature of the data shown in Figure 4 is nested within a U-shaped overall data pattern similar to that in Figure 3. These data (Figure 4) show a decline followed by an increase in training volume, with the undulating volume embedded within. Longtitudinal data may be a composite of several factors or signals occuring simultaneously. The coach and scientist can observe overall volume changes along the week-to-week changes within larger patterns and thereby establish a more nuanced understanding of how the training plan and athlete adaptation are unfolding [31,43].

Trend analyses (linear, curvilinear, cyclic) are useful methods for assessing the progress of various processes that occur during training. Unlike the measurement of minimum worthwhile changes, such as the differences between BP and TP and related protocols, trends provide a richer and more complete idea of how training interventions proceed. On a deeper level, there are many similarities between the analysis of minimum worthwhile changes and the analysis of trends. Trend analyses permit an investigator to track and observe how the training process unfolds and the potential influence of time-related factors along the historical path. As a time-series progresses forward in time, there is a practical need for determining how an athlete develops from moment-to-moment. The ability to analyze an athlete’s developmental process in near real-time is the type of approach that can assist in training management of the elite athlete and is the method to which we now turn.

## 4. Statistical Process Control (SPC)

Most individuals in athletics would agree that training is a process. Until recently, training management was largely a ‘seat-of-the pants’ endeavor, relying almost exclusively on the experience, knowledge, and personal philosophies of a coach or team of coaches. Statistical process control (SPC) uses continuous data sampling of a process to expose the patterns evident in the process. Originally, the idea was to measure how industrial and business processes of production could be measured and managed. SPC is a subset of Total Quality Management (TQM), which uses SPC and other management methods to measure and control nearly all aspects of a production process. SPC was implemented by W.E. Deming to assist Japan in industrial rebuilding after World War II [81,82]. SPC can also be used for research purposes [31,83,84], but its greatest strength lies in monitoring change and detecting outliers. SPC cannot identify the nature of the problem, only that something unusual and worthy of investigation has occurred or is occurring. Training *is a process* that could profit from many of the tenets of SPC. Continuous sampling of a single athlete’s training is synonymous with training monitoring.

Perhaps the most powerful technique of SPC that will find immediate use with elite athletes is the control chart [83,84,85,86]. Control charts are used to measure and monitor a process as it occurs. The primary information gained from control charts is the identification of trends, shifts, and outlier data. Outlier data in SPC refers to unusual data values that do not fit within the normal variability ranges of the process [58,59,87,88,89,90,91]. 

Control charts are the primary means of displaying the athlete’s training process over time. For example, if PPRH is collected systematically each day, one should expect some variability in the values obtained (Figure 5). Some variation is ‘normal’ (i.e., expected) and demands no explanation. Other variation can be considered abnormal or rare and because of the rarity merits further investigation. Athletes vary in their status and condition during all training durations, minute-to-minute, day-to-day, month-to-month, and year-to-year. By understanding an athlete’s performance and training variations, one gains the ability to determine when variation or change is too large to be expected by chance through normal variability. For example, if one plots the PPRH over time in conjunction with the mean and multiples of the standard deviation, the distribution of the PPRH values will vary predictably relative to the normal distribution. Figure 5 shows PPRH data from a former elite female gymnast over an entire collegiate season. Figure 5 is a control chart with information embellishments to help show how SPC works. 

A variable that is reflective of a process should indicate whether the process is ‘in’ or ‘out’ of control (i.e., varying normally or excessively). The process of athlete training should be in control, and variables reflective of that control should vary within relatively small ranges. As described above, human performance varies in many ways, but when plotted such as in Figure 5, the variability should remain within certain limits, defined by probability, in order to be considered in control. These limits are called ‘control limits,’ and although these limits can be calculated in a number of ways [81,82,83,84], assumptions of statistical normality are often used. As such, the distribution of data collected over time can be plotted and inspected horizontally. Figure 5 shows the normal distribution of the plotted data with normal curves (dotted lines) graphed on top of the longitudinal data for illustrative purposes. Calculated control limits (1.5 times and 2.0 times the SD) are also shown. Since the normal distribution is well understood, we can use the properties of the normal distribution to study the variability of PPRH in terms of statistical regularity or rareness. 

The athlete began training on 11 September and was immediately confronted with high volume training after a summer of little physical training. The early PPRH data met or exceeded the upper control limits four times, and most of the remaining data values were above the mean. These data may indicate that the athlete was facing unaccustomed training loads and that she is being loaded above her accustomed norm. The outlier data may or may not indicate something worth investigating. In this case, the athlete’s response should be expected to show higher than normal resting heart rates.

Figure 5 shows four PPRH values above +2.0 × SD and four below −2.0 × SD relative to the control limits. In terms of the +1.5 × SD control limit, the percentage of data values expected between the +1.5 × SD and −1.5 × SD control limits is approximately 93.3%. Thus, the probability of a data value greater or less than the +1.5 × SD control limit is approximately 6.7%, still indicative of a rare result meriting investigation. The ±1.5 × SD value has been chosen based on experience. Two standard deviations is often too high, making the detection of outlier values too restrictive [31]. Since our interest lies in excessively high PPHR values, we can note that four values met or exceeded the +2.0 × SD control limit, and 11 PPHR values exceeded the 1.5 × SD control limit (including those greater than 2.0 × SD). 

### Combining Data Histories with SPC

With illness considered a threat to safe and productive training, Figure 6 references the assessment with the use of SPC and two data series recorded by a monitoring tool, daily PPRH and the sum of illness symptoms (ILL) for a former elite gymnast [31,69,87,91]. Although there may not be a sufficient number of records to meet the criteria described for detecting differences and trends as discussed above, SPC identified data outliers (unusual or rare values) compared to normal variation. These results provided the coach, scientist, and physician with a systematic approach regarding the potential relationship between PPRH and illness, possibly enabling prediction of the onset of illness.

Although the ILL does not always reach a control limit for the number of illness symptoms, this analysis of ILL is limited by the use of a variable that can only assume a few values based on the presence or absence of illness symptoms, while resting heart rate can assume many values. The elegant simplicity of this approach for both variables remains helpful to the training and performance stakeholders in identifying outliers and their relationships. In this presentation, the number of variables is relatively small (i.e., two) and could be enhanced by including other variables from monitoring information.

SPC can lend considerable interpretative structure and value. The ability to identify illness periods and their relative influence on overall adaptation may serve as a window on the processes of change that the athlete is experiencing. Such information may serve as an evidence-based aid for enhanced decision making.

SPC combines all the techniques and methods described above and puts them to use in a process-monitoring format that can provide important information with minimum delay. The detection of outliers is important as “red flags” for coaches, scientists, and physicians. After all, if something is behaving normally, there is little need for concern or intervention. The need for concern arises when something gets “weird.” SPC uses probabilities to assess when something happens that deserves attention.

## 5. Conclusions

Studying and managing elite athletes can be extremely challenging because of their rareness in the athletic population and the sometimes suboptimal approach of applying traditional statistics to identify differences, trends, and outlier behavior when there are only one or a few elite athletes. Training and performance are staggeringly complex; the use of the methods described above helps reduce the universe of possible interpretations to a manageable level. Of course, the methods described here are not exclusively for elite athletes and can serve as a tool for any investigator who is interested in one or a few important individuals, or has the ability to process large amounts of individual athlete data. SPC in particular is a relatively simple method, intimately tied to training monitoring, that can help identify a process threat (i.e., training or performance), by indicating whether the threat is worth considering. SPC methods can be applied for near real-time measurement and management, unlike the most traditional approaches of waiting for a treatment’s influence to be observable or a trend to become manifest. It is hoped that this work can stimulate more sports scientists to understand, measure, and manage elite athletes. Considering the fact that “elite” sport is now a big industry, the impact of such research efforts could be large. 

## Figures and Tables

**Figure 1 sports-07-00105-f001:**
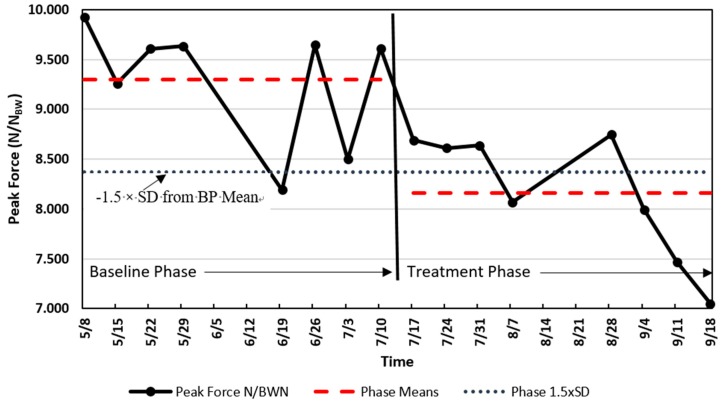
Single national team athlete analysis of scaled peak force (SPF) (N/N_BW_ [Newtons per Newtons Body Weight]) change. The solid line indicates the SPF, the dashed lines are the means of the two phases (baseline phase, BP, and treatment phase, TP), and the dotted line is the value of −1.5 times the standard deviation of the BP below the mean of the BP.

**Figure 2 sports-07-00105-f002:**
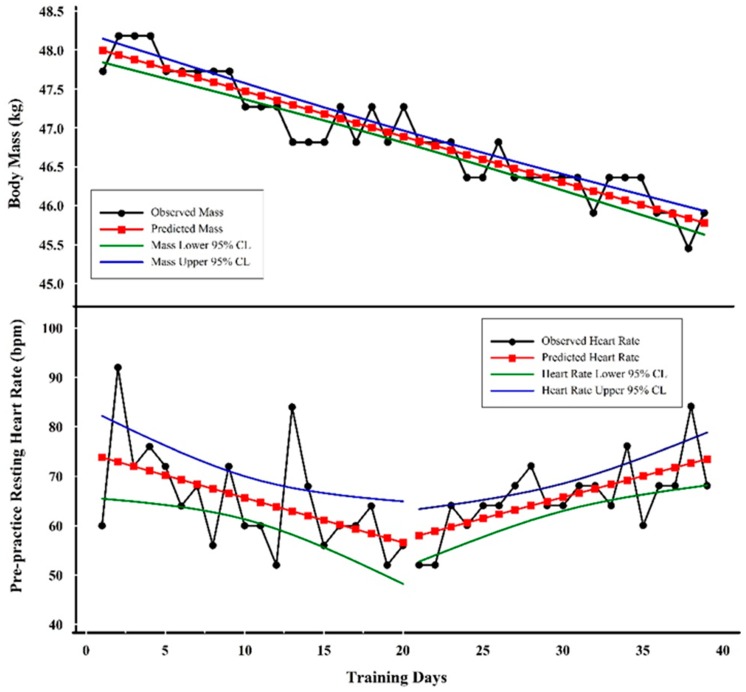
Examples of trend analyses with three linear trends. The top graph shows the linear decline in scale weight while the bottom graph shows an initial decline in pre-practice resting heart rate (PPRH) followed by an increase.

**Figure 3 sports-07-00105-f003:**
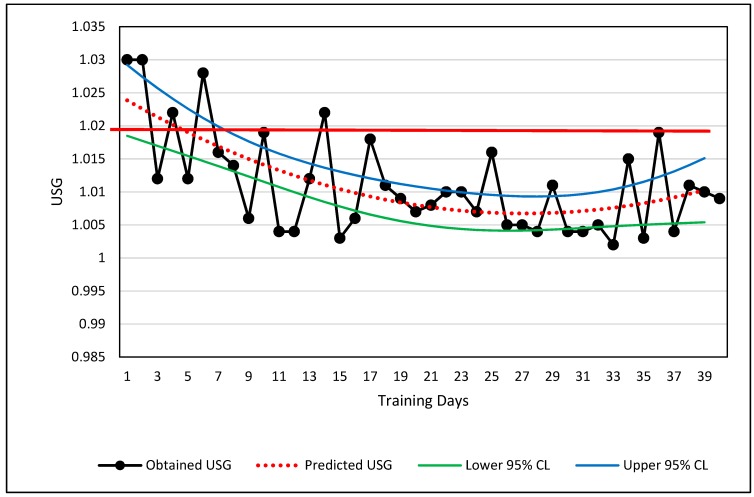
Curvilinear relationship of urine specific gravity across 3.5 months. The solid red line shows the value below which the athlete is considered adequately hydrated. USG measurements are shown via black lines and filled circles, the curvilinear regression line (dotted red line), and the upper and lower 95% confidence limits for the prediction of USG from knowing the day of training. A thicker red line is shown at 1.02 USG which serves as the boundary between adequately and inadequately hydrated.

**Figure 4 sports-07-00105-f004:**
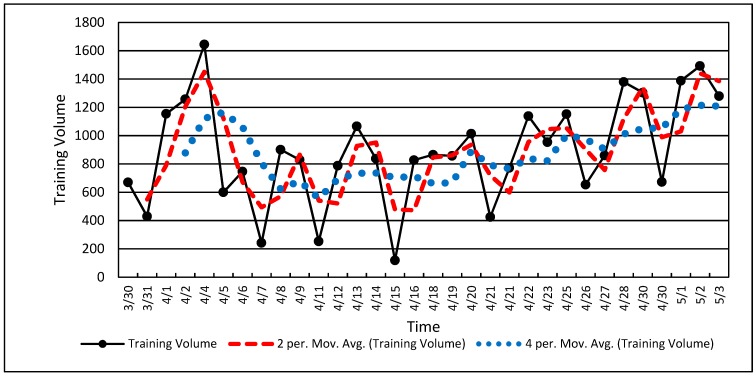
Cyclic single-athlete training volume data. Note the dotted line showing the two-point (two data points) and four-point (four data points) running average.

**Figure 5 sports-07-00105-f005:**
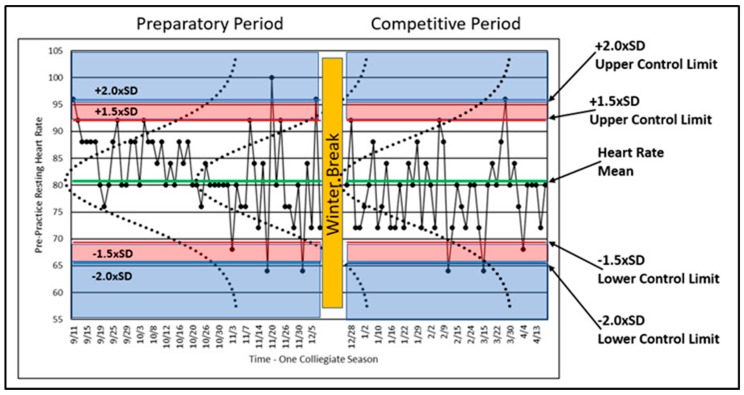
Control chart of a former elite gymnast’s PPRH for an entire collegiate season with associated mean, multiples of standard deviations, and normal curves.

**Figure 6 sports-07-00105-f006:**
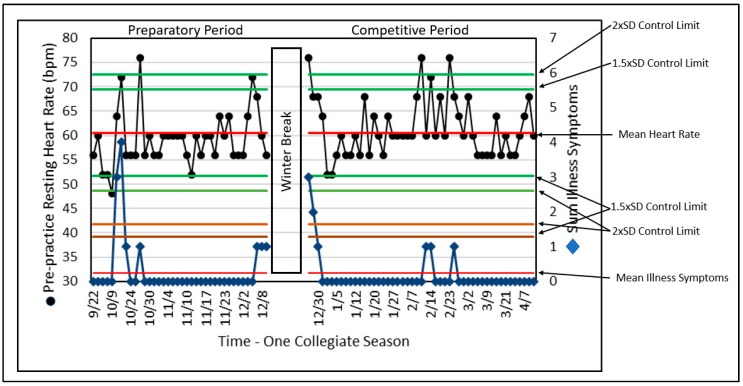
SPC of PPRH and the sum of illness symptoms (ILL) across a complete collegiate gymnastics season. Note that when illness symptoms are high there is a corresponding increase in PPRH.

**Table 1 sports-07-00105-t001:** Comparison of pre- and post-test data for a countermovement vertical jump.

Athlete	Pre-Test (cm)	Post-Test (cm)
01	34.20	40.65
02	**41.80**	**38.80**
03	30.60	35.90
04	29.50	34.36
05	**44.21**	**41.21**
06	34.80	39.80
07	35.10	40.40
08	**42.85**	**39.85**
09	31.55	38.50
10	29.10	34.50
Mean ± SD	35.35 ± 6.69	38.40 ± 2.56
Standard Error	1.80	0.81

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
