# Peer review of "Recommendations for Measurement and Management of an Elite Athlete"

_sports, 2019, doi:10.3390/sports7050105_

Round 1
Reviewer 1 Report
I think the article is interesting and may be published in the magazine. However, the theoretical framework should be structured in a better way, justifying the development of this research work
Author Response
Reviewer 1
I think the article is interesting and may be published in the magazine. However, the theoretical framework should be structured in a better way, justifying the development of this research work
Please note that I have revised the introductory material to include the following two paragraphs:
Studies of elite athletes are not rare, too often because the label “elite athlete” is applied too generously. Since 1974, the PUBMED online database records of a total of 6661 studies averaging 155 studies per year through 2018 using the search term “elite athlete.” The majority of studies of “elite athletes” have occurred since the year 2000 with a total of 5934, averaging approximately 312 studies per year (PUBMED accessed 6 April 2019). Measurement and management of real elite athletes is not a simple problem of categorization but one of acquiring reliable and valid data that can benefit all the stakeholders.
Traditional approaches to studies of sport and exercise science are built upon assumptions that rarely hold for the elite athlete. We believe that reframing elite athlete research and management will serve all of the stakeholders that pursue the highest levels of performance. The methods described below may also benefit any investigator, coach, or scientist who is faced with the problem of ensuring that training and performance are optimized for any single athlete.
Reviewer 2 Report
Please rearrange the abstract and the article because they are many spaces between phrases. Also, rearrange the tables and figures.
Please delete the phrase with blue color, see lines 388-399.
Line 529 - please rearrange the phrase
Please present the conclusions more broadly and please include the references.
Author Response
Reviewer 2
Please rearrange the abstract and the article because they are many spaces between phrases. Also, rearrange the tables and figures.
The modified abstract is below. Tables and Figures have been edited for position within the text.
Modified Abstract
Abstract: Athletes who merit the title ‘elite’ are rare and differ both quantitatively and qualitatively from athletes of lower qualifications. Research involving elite athletes suffers because of the typical nomothetic demands for large sample sizes and other statistical assumptions. Ideographic research uses single-athlete designs, trend analyses, and statistical process control. Single-athlete designs seek to measure differences in repeated measurements under prescribed conditions and trend analyses may expand understanding of the athlete and provide opportunities for prediction of future outcomes from the real world of actual training and competition. Statistical process control uses control charting and other methods that have scarcely been applied to athletes and may open an avenue for business-like control of the process of training and competition. This document provides examples of these types of analyses using all of these techniques with elite athlete data.
Old Abstract
Abstract: Athletes who merit the title ‘elite’ are rare and differ both quantitatively and qualitatively from athletes of lower qualifications. Research involving elite athletes suffers because of the typical nomothetic statistical demands for large sample sizes. Ideographic research uses single-athlete designs, trend analyses, and statistical process control. Single-athlete designs seek to measure differences in repeated measurements of single athletes under different conditions and trend analyses may expand understanding of the athlete and provide opportunities for prediction of future outcomes. Moreover, traditional probability-based assessments may provide all stakeholders with measured confidence in treatments, conditions, and outcomes. Statistical process control uses control charting and other methods that have scarcely been applied to athletes may open an avenue for business-like control of the process of training and competition. This document provides examples of these types of analyses using all of these techniques with elite athlete data.
Please delete the phrase with blue color, see lines 388-399.
I can’t seem to find any “blue color” in the manuscript.
Line 529 - please rearrange the phrase
The revised sentence is shown below:
A running average is a simple average of two or more data values obtained via successive calculations or iterations through the entire data set (Figure 4).
Please present the conclusions more broadly and please include the references.
The revised conclusion is shown below and references are included with the main manuscript.
Studying and managing elite athletes can be extremely challenging because of their rareness in the athletic population and the sometimes-suboptimal approach of applying traditional statistics to identify differences, trends, and outlier behavior when there are only one or a few elite athletes. Training and performance are staggeringly complex, and the ability of a stakeholder to identify and systematically enhance training and performance can be overwhelmed by logistics and other idiosyncratic issues. Of course, the methods described here are not exclusively for elite athletes and can serve as a tool for any investigator who is interested in a one or a few important individuals. SPC in particular is a relatively simple method, intimately tied to training monitoring, that can help identify a process threat (i.e., training or performance), by indicating whether the threat is worth considering. SPC methods can be applied for near real-time measurement and management, unlike the most traditional approaches of waiting for a treatment’s influence to be observable or a trend to become manifest. It is hoped that this work can stimulate more sports scientists to understand, measure, and manage elite athletes. Considering the fact that “elite” sport is now a big industry; the impact of such research efforts could be large.
Round 2
Reviewer 1 Report
The manuscript presented is very interesting and is a subject that has not been studied because of the difficulty of the sample size. I congratulate you. However, the document is a mixture of unordered ideas. They must reorder the document and establish which ideas are principal and which are not.
Author Response
I have reordered and rewritten portions of the manuscript to comply with what I believe to be Reviewer 1's criticisms. Sadly, neither the co-authors nor I have been able to understand the desires of Reviewer 1, but I've done my best to comply. I've placed the elite athlete discussions into the first section and follow that with why and how to analyze single athletes. Please forgive my lack of insight.
b
Reviewer 2 Report
Line 727 - Where is the title of figure, please insert
Author Response
I'm sorry that I cannot find the contentious missing title. Line 727 in the revised document reads as follows:
fitness acquisition. A 1-year study. J. Strength Cond. Res. 1995, 9, 110-115.
However, I've also checked all figures for a title and they all appear to have one.
Round 3
Reviewer 1 Report
The contributions of the authors are not enough, as the authors say the name elite athlete is too general, and the study and management of them is conditioned by the elite sport they execute. My comments are in this respect, should differentiate an athlete who competes individually with respect to an athlete who competes in a team, because in the latter the management of the team for the final result is very important. The authors must make reference to this aspect in their document, or modify the title of the article, since their document is not applicable to the whole concept of elite athlete
Author Response
Sentence 1
The contributions of the authors are not enough, as the authors say the name elite athlete is too general, and the study and management of them is conditioned by the elite sport they execute.
Not enough of what? What we indicated in the paper is exactly the opposite of this sentence. The “name elite athlete” is fine, but the approaches used to measure and manage elite athletes are specialized and unlike the methods used for lower level athletes to the extent that they demand further elucidation – thus, this paper.
Sentence 2
My comments are in this respect, should differentiate an athlete who competes individually with respect to an athlete who competes in a team, because in the latter the management of the team for the final result is very important.
The article doesn’t differentiate between team sports and individual sports because athletes and their measurement and management (if they’re elite athletes) do not demand anything different from that of individual sport athletes. This is one of the statements that led me to believe that the reviewer doesn’t understand the paper, the role of athlete measurement, and so forth. Teams are combinations of individuals. That is not to say that teams don’t involve some additional and different concepts in their development, but these issues are not the point of this paper. Teams may cook a collective meal, but they cannot digest it in a collective stomach. Whatever is done to a group of people is by definition applied to the individual members of that group. Team spirit, culture, grit, mental toughness and so forth are only measurable in the broadest sense. Trying to analyze these concepts turns into a slightly more sophisticated form of storytelling – not science. Qualitative methods have evolved to delve into these areas, but again, this was not the focus of our paper.
Sentence 3
The authors must make reference to this aspect in their document, or modify the title of the article, since their document is not applicable to the whole concept of elite athlete
We believe that the article is in fact applicable to the concept of elite athlete.